

# Auditory ERP response to successive stimuli in infancy

Ao Chen[1,2,3], Varghese Peter[1] and Denis Burnham[1]

[1] The MARCS Institute for Brain, Behaviour and Development, Western Sydney University, Penrith, Australia
[2] Institute of Linguistics, Utrecht University, Utrecht, Netherlands
[3] Department of Psychiatry, Brain Center Ruldolf Magnus, University Medical Center Utrecht, Utrecht, The Netherlands

## ABSTRACT

**Background.** Auditory Event-Related Potentials (ERPs) are useful for understanding early auditory development among infants, as it allows the collection of a relatively large amount of data in a short time. So far, studies that have investigated development in auditory ERPs in infancy have mainly used single sounds as stimuli. Yet in real life, infants must decode successive rather than single acoustic events. In the present study, we tested 4-, 8-, and 12-month-old infants' auditory ERPs to musical melodies comprising three piano notes, and examined ERPs to each individual note in the melody. **Methods.** Infants were presented with 360 repetitions of a three-note melody while EEG was recorded from 128 channels on the scalp through a Geodesic Sensor Net. For each infant, both latency and amplitude of auditory components P1 and N2 were measured from averaged ERPs for each individual note. **Results.** Analysis was restricted to response collected at frontal central site. For all three notes, there was an overall reduction in latency for both P1 and N2 over age. For P1, latency reduction was significant from 4 to 8 months, but not from 8 to 12 months. N2 latency, on the other hand, decreased significantly from 4 to 8 to 12 months. With regard to amplitude, no significant change was found for either P1 or N2. Nevertheless, the waveforms of the three age groups were qualitatively different: for the 4-month-olds, the P1–N2 deflection was attenuated for the second and the third notes; for the 8-month-olds, such attenuation was observed only for the middle note; for the 12-month-olds, the P1 and N2 peaks show relatively equivalent amplitude and peak width across all three notes. **Conclusion.** Our findings indicate that the infant brain is able to register successive acoustic events in a stream, and ERPs become better time-locked to each composite event over age. Younger infants may have difficulties in responding to late occurring events in a stream, and the onset response to the late events may overlap with the incomplete response to preceding events.

Corresponding author
Ao Chen, irischen71@hotmail.com

## INTRODUCTION

Electrical responses in the brain to external or internal events are called Event-Related Potentials (ERPs). While various behavioural methods are available, ERPs are especially

useful for studying auditory development early in infancy. Behavioral methods measure infants' auditory perception indirectly, using for example, looking time as a proxy for auditory attention; but as active attention is required, such methods are limited by infants' attentional capacity and so typically only relatively small amounts of data can be collected from each infant. In contrast, auditory ERP data collection does not require active participation, and a fairly large amount of data can be obtained within a short time period, which makes it a useful tool for studying auditory development in the early years of life (*Dehaene-Lambertz & Baillet, 1998*; *He, Hotson & Trainor, 2007*; *Kushnerenko et al., 2002*; *Trainor, 2012*).

Auditory ERPs mature from birth to childhood. The usual peaks in adult auditory ERPs, P1 (positive deflection between 40–60 ms after stimulus onset), N1 (negative deflection between 90–110 ms after stimulus onset), P2 (positive deflection between 140–170 ms after stimulus onset), N2 (negative deflection between 220–280 ms after stimulus onset), may not be visible in infant auditory ERPs (*Barnet et al., 1975*; *Ceponiene, Cheour & Näätänen, 1998*; *Kushnerenko et al., 2002*; *Näätänen, 1992*; *Ponton et al., 2000*; *Wunderlich, Cone-Wesson & Shepherd, 2006*). When presented with stimuli at inter-stimulus intervals (ISIs) shorter than 1000 ms, the auditory ERPs of infants and young children exhibit a positive peak around 150 ms (P1) after stimulus onset, a negative peak around 250 ms (N2), another positive peak at around 350 ms (P2), and another negative peak at around 450 ms (N3) (*Ceponiene, Cheour & Näätänen, 1998*; *Kushnerenko et al., 2002*). Many studies have found decreases in response latency as infants mature (*Barnet et al., 1975*; *Ceponiene, Cheour & Näätänen, 1998*; *Kushnerenko et al., 2002*; *Näätänen, 1992*; *Ponton et al., 2000*; *Wunderlich, Cone-Wesson & Shepherd, 2006*), probably due to increasing neuronal myelination over age (*Eggermont & Salamy, 1988*; *Moore & Guan, 2001*). Yet, such reductions of latency are not found for all peaks. *Little, Thomas & Letterman (1999)* tested 5- to 17-week-old infants with 100 ms duration tones as well as clicks, and found latency decrease over age for the late peaks (N2, P2), but not for the earlier P1 (peak around 150 ms after the stimuli onset). However, testing older infants longitudinally at 3, 6, 9, and 12 months on three 100 ms harmonic tones, *Kushnerenko et al. (2002)* found that P1 latency decreased as the infant grew older, whereas N2 latency did not show significant difference across ages. The different patterns in these two studies suggest a non-linear development of peak latency, i.e., the latency decrease for certain peaks might be more evident within a specific age window. In addition, *Little, Thomas & Letterman (1999)* used only one tone (400 Hz) whereas *Kushnerenko et al. (2002)* used three tones (500 Hz, 625 Hz, and 750 Hz) and the ERPs were obtained by averaging the responses to all the tones. These different stimuli may have influenced developmental patterns.

The development of the amplitude of these auditory ERP peaks shows an even more complex picture. *Barnet et al. (1975)* tested infants from 10 days to 37 months, using clicks as stimuli, and found that P1–N2 deflection was the "landmark" to measure auditory ERP, as it is present in all participants, and the P1–N2 amplitude increased over age. In the *Little, Thomas & Letterman (1999)* study stimulus specific developments of P1 and N2 amplitudes was found: P1 showed a significant increase for clicks but not tones;

N2 showed a quadratic trend for both the tones and clicks, but in the opposite direction. For the tones, N2 amplitude first increased and then decreased from 5 to 17 weeks, whereas the opposite was true for the N2 of the clicks. *Kushnerenko et al. (2002)* demonstrated that P1 amplitude at birth was significantly smaller than at older ages, and remained stable from 3 months to 12 months, and that N2 increased in amplitude (became more negative) between 3 and 9 months. *Jing & Benasich (2006)* found that N2 and P3 increased between 3 and 9 months, and then decreased with age. Together these findings suggest that the development of auditory ERPs is highly stimulus specific. Similar to latency, amplitude change might be more evident within a specific age window for certain peaks. Nevertheless, the P1–N2 deflection seems to be a reliable marker for the auditory onset response.

In these studies, quite often single sounds (e.g., clicks or single tones) have been used as stimuli. Yet in real life, we often need to decode successive rather than single acoustic events. For example, speech consists of multi-word utterances (e.g., *van de Weijer, 1998*), and the segmentation of words from the continuous speech stream is fundamental to infants' language learning. Similarly, music is composed of multi-note bars and phrases, often without pauses between notes (*Krumhansl & Jusczyk, 1990*). Efficient on-line processing of real life auditory input requires efficient accurate processing of these rapidly presented successive signals. Impairments in such successive processing are associated with reading and language impairments (*Tallal, 2004*) and are predictive of later language skills when assessed in infancy (*Choudhury & Benasich, 2011*). Therefore, understanding how the infant's brain responds to successive stimuli is fundamental to studying high level processing. Mismatched negativity (MMN), the neural detection of regularity violation (see *Näätänen et al., 2007*; *Paavilainen, 2013*), is a method that has been widely used to understand human auditory discrimination. An MMN is elicited when listeners encounter infrequent sounds embedded in repeating frequent sounds. Multi-tone melodies have been used as stimuli to understand infants' mismatched responses (*Stefanics et al., 2009*; *Tew et al., 2009*), yet these studies often incorporate relatively long inter-tone intervals between the notes composing the melodies. For example, the tones used by *Stefanics et al. (2009)* were of extremely short duration (50 ms), with a relatively long inter-tone intervals (150 ms) and inter-pair intervals much longer than the tones themselves (1250 ms).

In real life, however, sounds tend to have a much longer duration and occur successively with much shorter inter-sound intervals, if at all. Speech syllables in running speech tend to have a duration of around 150–200 ms (*Koopmans-van Beinum & van Donzel, 1996*), while musical tempi tend to cluster between 90 and 120 beats per minute, i.e., each beat is longer than 500 ms, and often contains multiple notes (*Moelants, 2002*). Previous findings have shown that infants and adults respond to fast versus slow temporal modulations differently (*Telkemeyer et al., 2009*; *Giraud & Poeppel, 2012*), hence it is important to investigate how infants' brain responds to sound streams that represent real life characteristics, as this provides the basis for understanding higher level neural function. Given that ERPs develop in a stimulus-specific manner (*Little, Thomas & Letterman, 1999*; *Wunderlich, Cone-Wesson & Shepherd, 2006*), it is important to gain

insight into how the infant brain registers real life successive acoustic events, so that higher level processing such as MMN can be better understood. In the current study, we used a melody comprised of three successive piano notes, and examined the development of auditory ERPs, specifically the infantile auditory onset responses, the P1 and N2. We tested 4-, 8-, and 12-month-old infants, and investigated how P1 and N2 latency and amplitude change across the three tones, and how P1 and N2 to successive tones develop in the first year of life. Auditory ERPs reflect neural firing time-locked to the stimuli, and exhibit high temporal resolution. As the P1–N2 onset complex is generated with a longer latency among young infants, we suggest that that upon encountering successive acoustic events, young infants may not be able to process each event sufficiently before the next one is initiated. Older infants, on the other hand, may be able to register the sounds much more rapidly, and hence may be more capable of processing each successive tone in the 3-tone stimulus.

## METHODS

### Ethics

The ethics committee for human research at Western Sydney University approved all the experimental methods used in the study (Approval number: H9660). Informed consent was obtained from parents of all the participants.

### Participants

Eighteen 4-month-old (range: 4 month 4 days-5 month 5 days, 10 girls), 18 8-month-old (range: 7 month 22 days-9 month 4 days, 10 girls), and 18 12-month-old (range: 11 month 10 days-13 month 3 days, 11 girls) healthy infants participated in the study. All the infants were raised in a monolingual or predominantly Australian English environment. The current study is part of larger project, where we compare the perceptual development of musical versus speech stimuli. We selected these three age groups because the infants are assumed to tune into the sound structure of their native language between 4 and 12 months. None of the parents reported any hearing impairment or any ear infection within the two weeks before the experiment. One 12-month-old girl was tested but excluded from the analysis as she pulled the EEG cap off shortly after the experiment started, and one 8-month-old boy was excluded due to fussiness.

### Stimuli

Piano tones F3 (174.61 Hz), G3 (196 Hz) and A#3 (223.08 Hz) were synthesized in Nyquist, where the frequency of middle A (A4) was the usual 440 Hz. The notes were generated with the default duration of one-16$^{th}$ of a note (250 ms). Then, they were concatenated to form a rising melody. In order to ensure continuity and naturalness, the whole melody was then adjusted in Praat (*Boersma & Weenink, 2013*) using the overlap-add method to a duration of 667.5 ms, which resulted in slightly different durations for each note–first note duration = 220 ms, second note duration = 227 ms, and third note duration = 221 ms. The stimuli were presented through two audio speakers (Edirol MA-10A) each 1 m from the infants, and each 30 cm from the midline of the infant's

sagittal plane. The stimuli were presented at 75 dB SPL, with a random inter-stimulus interval varying between 450 and 550 ms. The stimuli were played through Presentation 14.9 (Neurobehavioral Systems).

## Procedure

The infants sat on the caregiver's lap in a sound attenuated room. An infant friendly movie was played silently on a screen ~1 m from the infants (in between the two audio speakers) to keep them engaged. Parents were instructed not to talk during the experiment. A maximum of 360 trials were presented, but the experiment terminated if the infant became restless. The total duration of the recording was around 7 minutes. EEG was recorded from 128 channels on the scalp through a Geodesic Sensor Net (net sizes: 40–42 cm, 43–44 cm, or 44–47 cm, depending on the head size of the infants), and all electrodes had impedance lower than 50 kΩ at the beginning of the experiment. The EEG was recorded at a sampling rate of 1000 Hz, and the online recording was referred to the vertex.

## EEG analysis

The EEG was analyzed offline using NetStation 4.5.7 software and EEGLAB toolbox (version 13.1.1b) in Matlab (2011b). The raw recordings were filtered in NetStation 4.5.7 between 0.3–20 Hz. The filtered recordings were down-sampled to 250 Hz before further analysis. The continuous recordings were segmented into 1000 ms epochs from the 100 ms before the onset (baseline) to 900 ms after the onset of the first note. The 15 electrodes at the peripheral positions of the net were removed from the analysis due to contamination from muscle movements. Next, for each participant, bad channels were identified by visual inspection and were interpolated (mean number of interpolated channels = 4, SD = 2). EEG was then re-referenced to an average reference. As it was impossible for the infants to sit still during the entire experiment, ERPs recorded from the parietal and occipital electrodes were contaminated by head movements. Auditory ERPs are mainly central-frontally distributed, hence conducting artifact reduction on all channels would waste clean signal from the central-frontal site. In order to sufficiently remove artifacts while retaining sufficient data from each child, we conducted artifact rejection on the 25 frontal channels (Fig. 1), where trials having an amplitude larger than 150 microvolts were removed. The remaining artifact free trials were averaged to obtain the ERP values for each infant. After artifact removal, the 4-month-olds had a mean of 248 trials (standard deviation (SD) = 39.87) accepted, the 8-month-olds a mean of 228 trials (SD = 63.64) accepted, and the 12-month-olds a mean of 275 trials (SD = 38.1) accepted. Channel 5, 11 and 12 were averaged to represent frontal central scalp ERP responses (FC), which correspond to the location of Fz on a 10–20 system. Figure 1 indicates the 25 channels used for artifact reduction and the channels averaged to represent FC. For each age, a grand average ERP was computed by averaging the ERPs of all participants.

The P1s for each of the three notes were detected in the windows 0–200 ms, 200–400 ms, and 500–700 ms respectively from the onset of the melody. The P1 was defined as the

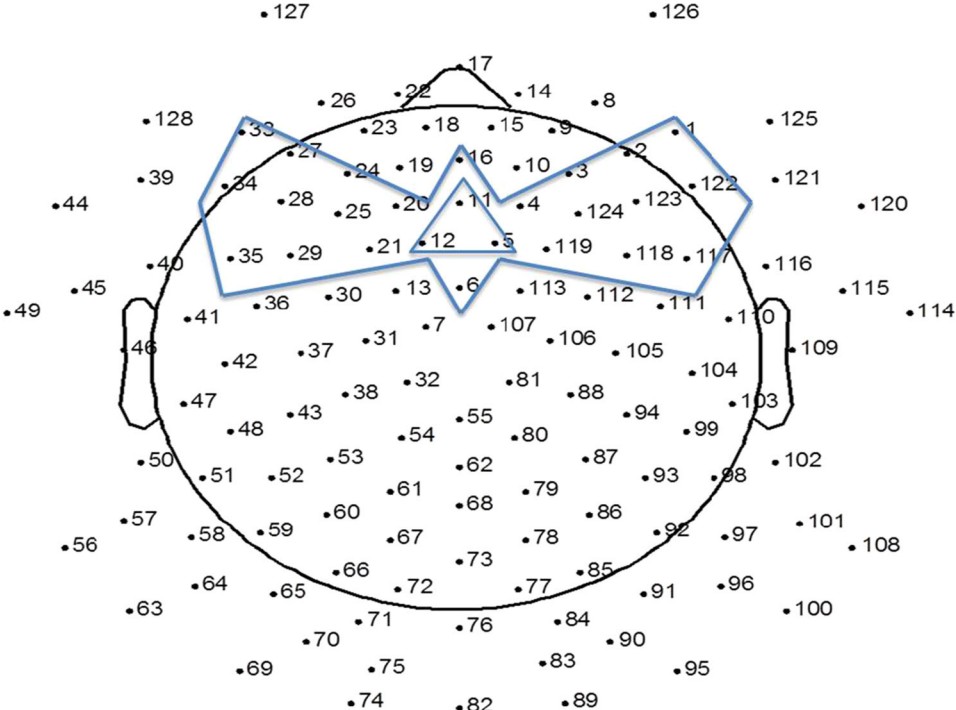

**Figure 1** The 25 channels used for artifact reduction (the area circumscribed by the bow-tie shape) and the three channels averaged for representing response at FC (circumscribed by the triangle shape).

highest positive peak in the above time windows. The N2s to each note were detected in the windows 200–300 ms, 400–600 ms, and 600–700 ms respectively from the onset of the melody. N2 was the lowest negative peak in the above time windows. Next, for each individual participant, the P1 and N2 peaks for each note were identified in the ±25 ms window of the corresponding grand average peaks.

## RESULTS

Figure 2 plots the grand average ERPs for the three age groups across time. As can be seen, the onset responses for each note, namely the infantile P1 and N2, were easily identifiable at each age.

Table 1 shows the latency of P1 and N2 for each note at FC. For each note separately, the latency measurements of P1 and N2 at FC were submitted to a univariate ANOVA, with age as the between-subjects variable. When age showed a significant result, Bonferroni-corrected post-hoc tests were conducted for pair-wise comparisons. Table 2 shows the ANOVA results for the latency measurements.

These results demonstrate that in general, the latency of ERP peaks is lower for older infants. More specifically, reduction of P1 latency is mainly observed between 4 and 8 months, whereas the latency of N2 decreases monotonically and consistently from 4 to 8 to 12 months. It can be observed that at the onsets of the second and the third note,
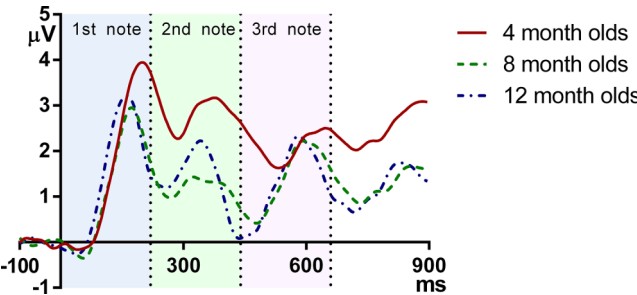

**Figure 2** **Grand average responses of the three age groups.** The vertical axis indicates the onset of the melody, and the vertical dotted lines indicate the onsets of the second and the third note.

**Table 1** **Mean P1 and N2 latency of each note with regard to the onset of the melody separated by age groups. Standard deviations are given in parentheses.**

| ERP peaks | 1st note | 2nd note | 3rd note |
|---|---|---|---|
| 4 m P1 | 195.78 (16.49) | 375.55 (20.86) | 648.89 (18.59) |
| 4 m N2 | 288.44 (17.35) | 532.89 (19.43) | 722.89 (19.30) |
| 8 m P1 | 169.41 (13.34) | 325.88 (20.84) | 587.76 (17.00) |
| 8 m N2 | 268.94 (20.76) | 470.12 (18.72) | 732.00 (17.83) |
| 12 m P1 | 162.12 (16.68) | 339.53 (16.36) | 581.65 (12.50) |
| 12 m N2 | 250.12 (19.60) | 439.76 (19.60) | 712.14 (17.41) |

**Table 2** **Effects of age for P1 and N2 latency measurements for each note.**

| | Effect of age | Post-hoc test |
|---|---|---|
| 1st note P1 | $F_{(2, 49)} = 22.72$, $p < 0.01$ | 4 m v.s. 8 m, $p < 0.01$ |
| | | 4 m v.s. 12 m, $p < 0.01$ |
| | | 8 m v.s. 12 m, n.s. |
| 1st note N2 | $F_{(2, 49)} = 17.33$, $p < 0.01$ | 4 m v.s. 8 m, $p < 0.05$ |
| | | 4 m v.s. 12 m, $p < 0.01$ |
| | | 8 m v.s. 12 m, $p < 0.05$ |
| 2nd note P1 | $F_{(2, 49)} = 30.2$, $p < 0.01$ | 4 m v.s. 8 m, $p < 0.01$ |
| | | 4 m v.s. 12 m, $p < 0.01$ |
| | | 8 m v.s. 12 m, n.s. |
| 2nd note N2 | $F_{(2, 49)} = 117.92$, $p < 0.01$ | 4 m v.s. 8 m, $p < 0.01$ |
| | | 4 m v.s. 12 m, $p < 0.01$ |
| | | 8 m v.s. 12 m, $p < 0.01$ |
| 3rd note P1 | $F_{(2, 49)} = 92.08$, $p < 0.01$ | 4 m v.s. 8 m, $p < 0.01$ |
| | | 4 m v.s. 12 m, $p < 0.01$ |
| | | 8 m v.s. 12 m, n.s. |
| 3rd note N2 | $F_{(2, 49)} = 5.01$, $p < 0.05$ | 4 m v.s. 8 m, n.s. |
| | | 4 m v.s. 12 m, n.s. |
| | | 8 m v.s. 12 m, $p < 0.01$ |

the 12-month-olds had a more negative deflection than the 4- and 8-month-olds, which suggests that the younger groups have a later complement of N2 than the older ones.

Next we examined the P1 and N2 latency with regard to the onset of each note. Figure 3 plots the P1 and N2 latency respectively with regard to note onset for each age group. Mixed design ANOVAs were conducted for P1 and N2 latencies with notes (Notes 1–3) as the within-subjects variable, and age as between-subjects variable. For P1, there was an overall significant main effect of note, $F(2, 98) = 122.49$, $p < 0.01$. Bonferroni-corrected post-hoc analysis indicated that P1 was significantly later for Note 1 than for Notes 2 and 3, both $p < 0.001$; and P1 of Note 2 was significantly earlier than both Notes 1 and 3, both $p < 0.001$. Age also showed a significant main effect, $F(2, 49) = 100.48$, $p < 0.01$. The 4-month-olds had a significantly later P1 compared to the 8- and 12-month-olds ($p < 0.001$) with no significant difference between the 8- and 12-month-olds. A significant interaction was found between notes and age, $F(2, 98) = 8.96$, $p < 0.01$. Bonferroni post-hoc tests indicated that: for the 4-month-olds, P1 of Note 2 was significantly earlier than that of Notes 1 and 3 (both $p < 0.01$) with no significant difference between Notes 1 and 3; for the 8- and 12-month-olds, P1 of Note 2 was significantly earlier than that of Notes 1 and 3 (both $p < 0.01$), and P1 for Note 1 was significantly later than that of Note 3 (both $p < 0.05$).

For N2, notes had a significant main effect, where $F(2, 98) = 10.15$, $p < 0.01$, and Bonferroni-corrected post-hoc tests indicated that the N2 of Note 2 was significantly earlier than that of Notes 1 and 3, with no significant difference between these two. Age was also significant, $F(2, 49) = 58.83$, $p < 0.01$. Bonferroni-corrected post-hoc tests indicated that the N2 latency decreased significantly from 4 to 8 months, and from 8 to 12 months, both $p < 0.05$. A significant interaction was found between notes and age, $F(2, 98) = 34.00$, $p < 0.01$, Bonferroni corrected post-hoc tests indicated that: for 4-month-olds, N2 latency of Note 2 was significantly later than that of Notes 1 and 3 (both $p < 0.01$); whereas on the other hand, for the 8- and 12-month-olds, N2 latency of Note 2 was significantly earlier than that of Notes 1 and 3, with N2 latency of Note 3 being significantly later than that of Note 1 (all $p < 0.05$).

In summary, P1 latency for each note decreases as the infants grow older. Interestingly, for all the three ages, the P1 of the middle note, Note 2, had the shortest latency compared to P1 of the edge notes. For the 4-month-olds, the P1 latency of Note 3 is longer than that of Note 1, whereas the 8- and 12-month-olds showed the opposite pattern. For the N2, the 4-month-olds had a longer N2 latency for Note 2, whereas for the other two groups, N2 latency was the shortest for Note 2.

Mean P1 and N2 amplitudes of each note for the three age groups are provided in Table 3 and Fig. 4 plots the mean P1 and N2 amplitude of each note of the three age groups. ANOVAs were conducted for the absolute amplitude of P1 and N2 with notes as the within-subjects variable and age as the between-subjects variable. For P1, a significant main effect of notes was found, $F(2, 98) = 9.01$, $p < 0.01$, but no significant main effect of age. Bonferroni-corrected pair-wise analyses indicated that P1 amplitude was significant larger at Note 1, compared to that at Notes 2 and 3 (both $p < 0.01$), with no difference between these two. For N2, notes had a marginally significant

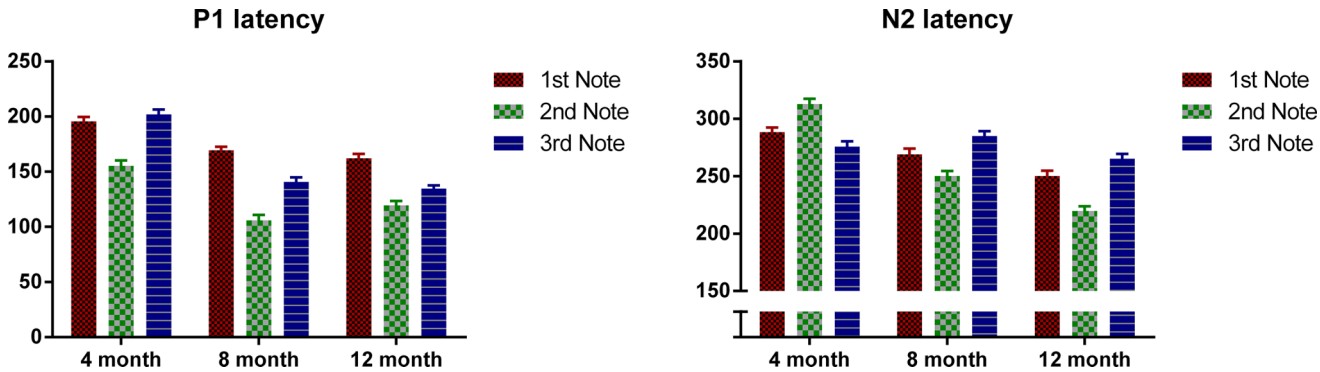

**Figure 3** Mean P1 and N2 latency with regard to each note's onset of the three age groups. Error bars represent standard errors.

main effect, F(2, 98) = 2.95, p = 0.057, and no significant difference was found between any pair of the three notes, and there was no significant main effect of age. For neither P1 nor N2, was there a significant interaction between age and notes. Similarly, for P1–N2 peak-to-peak amplitude, notes again showed a significant main effect, F(2, 98) = 3.66, p < 0.01; the P1–N2 peak-to-peak amplitude of Note 3 was significantly smaller than that of Note 1 (p < 0.05), whereas no significant difference was found between either Notes 1 and 2, or Notes 2 and 3. As notes had a significant main effect for all the three amplitude measurements, we examined the effect of age for each note separately. For both the P1 and N2 amplitudes, age did not show a significant effect for any of the notes, and no pair-wise difference was significant. For P1–N2 amplitude, no significant effect of age was found for either Notes 1 or 2, but for Note 3, there was a significant age effect, F(2, 49) = 3.45, p < 0.05; Bonferroni-corrected pair-wise analyses showed a marginally significant difference between the 4- and 12-month-olds, with the 4-month-olds having a smaller amplitude (p = 0.06).

In summary, P1 latency for each note decreases as the infants grow older. Interestingly, for all the three ages, the P1 of the middle note, Note 2, had the shortest latency compared to P1 of the edge notes. For the 4-month-olds, the P1 latency of Note 3 is longer than that of Note 1, whereas the 8- and 12-month-olds showed the opposite pattern. For the N2, the 4-month-olds had a longer N2 latency for Note 2, whereas for the other two groups, N2 latency was the shortest for Note 2.

Mean P1 and N2 amplitudes of each note for the three age groups are provided in Table 3 and Fig. 4 plots the mean P1 and N2 amplitude of each note of the three age groups. ANOVAs were conducted for the absolute amplitude of P1 and N2 with notes as the within-subjects variable and age as the between-subjects variable. For P1, a significant main effect of notes was found, F (2, 98) = 9.01, p < 0.01, but no significant main effect of age. Bonferroni-corrected pair-wise analyses indicated that P1 amplitude was significant larger at Note 1, compared to that at Notes 2 and 3 (both p < 0.01), with no difference between these two. For N2, notes had a marginally significant main effect, F(2, 98) = 2.95, p = 0.057, and no significant difference was found between any pair of the three notes, and there was no significant main effect of age. For neither P1

**Table 3 Mean P1 and N2 amplitude (in μV) of each note of each age group.** Standard deviations are given in parenthesis.

|            | 1st note    | 2nd note    | 3rd note    |
|------------|-------------|-------------|-------------|
| 4 m P1     | 4.09 (2.23) | 3.37 (2.56) | 2.77 (2.18) |
| N2         | 1.96 (2.34) | 1.50 (1.86) | 1.81 (1.99) |
| 8 m P1     | 3.26 (2.36) | 1.99 (2.81) | 2.78 (1.99) |
| N2         | 1.76 (2.15) | 0.51 (2.09) | 1.00 (1.76) |
| 12 m P1    | 3.53 (1.45) | 2.59 (1.67) | 2.70 (2.07) |
| N2         | 1.20 (1.47) | 0.09 (1.72) | 0.81 (2.71) |

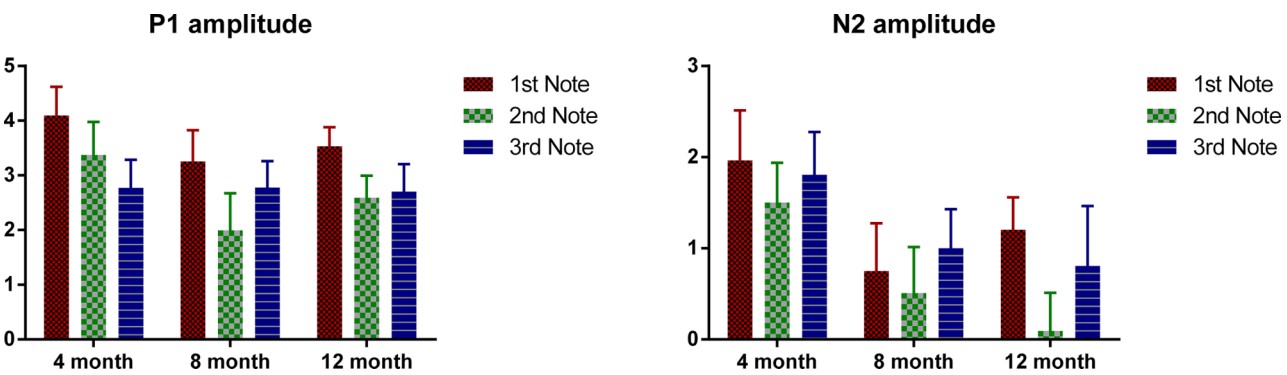

**Figure 4 Mean P1 and N2 amplitude of each note of each age group.** Error bars represent standard errors.

nor N2, was there a significant interaction between age and notes. Similarly, for P1–N2 peak-to-peak amplitude, notes again showed a significant main effect, $F(2, 98) = 3.66$, $p < 0.01$; the P1–N2 peak-to-peak amplitude of Note 3 was significantly smaller than that of Note 1 ($p < 0.05$), whereas no significant difference was found between either Notes 1 and 2, or Notes 2 and 3. As notes had a significant main effect for all the three amplitude measurements, we examined the effect of age for each note separately. For both the P1 and N2 amplitudes, age did not show a significant effect for any of the notes, and no pair-wise difference was significant. For P1–N2 amplitude, no significant effect of age was found for either Notes 1 or 2, but for Note 3, there was a significant age effect, $F(2, 49) = 3.45$, $p < 0.05$; Bonferroni-corrected pair-wise analyses showed a marginally significant difference between the 4- and 12-month-olds, with the 4-month-olds having a smaller amplitude ($p = 0.06$).

## DISCUSSION

In the present study, we tested 4-, 8- and 12-month-old infants on their auditory ERPs to a three-note melody. For all the three successive notes, there was an overall decrease in P1 and N2 latency over age. This indicates an increase in the speed of neural transmission, which is presumably related to the increased myelination of neurons over age. Latency decrease has been previously found in the response to single acoustic stimuli (e.g. *Barnet et al., 1975*; *Novak et al., 1989*; *Kushnerenko et al., 2002*), but as the results here show decreasing latencies to successive stimuli in a stream, this implies that infants are

able to segment a sound stream into its composite elements, with individual registration of each element at the neural level. Even at 4 months, although the N2 was not fully expressed when the next note occurred, P1 for the next note was still visibly evident. Nevertheless, as the onset response to the late events in a stream may overlap with the response from preceding events, the non-initial response is possibly more variable across participants. *Kushnerenko et al. (2002)* found a decrease in P1 but not in N2 latency from 6 to 12 months, and N2 was not visibly evident before 6 months. *Little, Thomas & Letterman (1999)* found a decrease in N2 but not in P1 from birth to 17 weeks. In the present study, we found consistent latency decrease across age for both peaks. These different findings are possibly due to the different stimuli: *Little, Thomas & Letterman (1999)* used one single harmonic tone, *Kushnerenko et al. (2002)* used three single harmonic tones, and we used a stream containing three piano tones. It is possible that the relatively rich spectral content of our stimuli allowed N2 to be visibly evident at a younger age than in the *Kushnerenko et al. (2002)* study.

Interestingly, the middle Note 2 elicited an earlier P1 latency compared to the edge Notes 1 and 3. The early P1 of the middle note is unlikely to be solely due to the incomplete response to the first note. If this were the case, then Note 3 should have had an equally early P1 peak. It seems that infant brain does not respond to the acoustic events at different positions of a stream in an identical manner. For the moment, it is difficult to ascertain what might cause the change in P1 latency across successive notes, yet it seems that the response to sounds at a central position in a stream tend to be more variable. In addition, for the 8-month-olds, the amplitude of the middle note seems to be more attenuated compared to the response to the first and third notes. Whether the neuronal processes are different for edge events and for medial events in a stream should be further tested, preferably with streams containing more than three notes. The majority words in English have a strong-weak pattern (*Cutler & Carter, 1987*), and whether such metrical structure might lead to a more prominent response to the onset event in an acoustic stream should be investigated by testing participants whose native language exhibits the opposite metrical pattern (e.g., Turkish).

With regard to amplitude, P1 was larger for the first than for the subsequent notes. Neither P1 nor N2 amplitude changed significantly across age, although it should be acknowledged that individual variation is large for amplitude measurements, and such variation may have masked any age group differences. Even though the statistical tests failed to show any consistent age effect, the grand average waveforms are qualitatively different for each age (see Fig. 2). For the 4-month-olds, clear P1s and N2s can be observed for Note 1 whereas those to the two subsequent notes are much more attenuated. For the 8-month-olds, compared to the edge notes, the central Note 2 has smaller P1 and N2 amplitudes. In addition, the P1 peak of the central note shows a plateau-like morphology, whereas the responses to the edge notes have sharper P1 peaks. The P1 and N2 peaks of the 12-month-olds show a comparably equivalent amplitude and peak width across all three notes. Thus in summary, compared to older 8- and 12-month-old infants, the younger 4-month-old infants had a smaller P1–N2 deflection for Note 3, even this was the final note, with no other acoustic event following it. This suggests that

young infants may have difficulties in responding to late occurring events in a stream. Such attenuated amplitude may be due to neural refractoriness (*Budd et al., 1998*; *Gilley et al., 2005*; *Ritter, Vaughan & Costa, 1968*; *Sable et al., 2004*; *Sussman et al., 2008*) with processing of late events being hindered by unfinished processing of preceding events. In other words, more time may be required for neurons to reset and respond to following acoustic events for young infants.

## CONCLUSION

In this study we examined auditory cortical responses to successive musical notes of infant at ages of 4, 8, and 12 months. Clear P1 and N2 peaks were identified for all three notes at all three ages, indicating successful neural registration of individual components in a continuous stream. While there was a significant decrease in P1 and N2 latencies all three notes as infant age increased, these results show, for the first time, individual neural registration of components of continuous multi-element events, even in relatively young infants. Over and above this general finding age-related nuances were evident: P1 and N2 amplitudes were larger for the first note in the stream than for subsequent notes, and additionally, for the final Note 3, the 4-month-olds had a smaller amplitude compared to the two older groups, suggesting that, despite individual registration, young infants may have greater difficulty registering late occurring events in a stream. This attenuated amplitude for late events at the younger age points to the possible nature of developmental progression in auditory event registration in infancy; there could well be neural registration of individual elements in an acoustic stream, but, possibly due to incomplete processing of each event, there may be cumulative reduction of processing resources within a certain refractory period. This would imply temporal and numerical limits on successive component event processing, which are possibly related to memory limits. Further studies of such limits and their reduction over age will greatly improve the understanding of speech perception and language development as well as the development of other complex abilities such as music and other event processing. For young infants, P1 and N2 latencies change over the course of the experiment. This study provides the basis for such future studies by showing, for the first time, that as young as 4 months, infants show specific neural registration of successive frequency-modulated events in an otherwise continuous stream, and continuing maturation of the brain response to successive sounds in the first year of life.

### Funding

This study was supported by Endeavor Research Fellowship funded by Australian Department of Education to the first author, with grant number ERF_PDR_113381_2013. The study was also supported by Discovery Project Grant funded by Australian Research Council to the last author, with grant number DP110105123. The funders had no role in

study design, data collection and analysis, decision to publish, or preparation of the manuscript.

## Grant Disclosures

The following grant information was disclosed by the authors:
Australian Department of Education: ERF_PDR_113381_2013.
Australian Research Council: DP110105123.

## Competing Interests

The authors declare that they have no competing interests.

## Author Contributions

- Ao Chen conceived and designed the experiments, performed the experiments, analyzed the data, contributed reagents/materials/analysis tools, wrote the paper, prepared figures and/or tables, reviewed drafts of the paper.
- Varghese Peter conceived and designed the experiments, analyzed the data, contributed reagents/materials/analysis tools, wrote the paper, prepared figures and/or tables, reviewed drafts of the paper.
- Denis Burnham wrote the paper, reviewed drafts of the paper.

## Human Ethics

The following information was supplied relating to ethical approvals (i.e., approving body and any reference numbers):

The ethics committee for human research at Western Sydney University approved all the experimental methods used in the study (Approval number: H9660).

## Data Deposition

http://dx.doi.org/10.4225/35/56450b59ad2c2.

## Supplemental Information

Supplemental information for this article can be found online at http://dx.doi.org/10.7717/peerj.1580#supplemental-information.

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
