# Peer review of "Auditory ERP response to successive stimuli in infancy"

_PeerJ, doi:10.7717/peerj.1580_

## Round 0.1 · original submission · Major Revisions

Dear Authors, This manuscript has received comments from three peer reviewers which requires that the manuscript undergo major revision. Please do the revisions required and resubmit it so that the same reviewers can review again the repairs and improvements made to the manuscript based on the comments.

Thanking You.

·

Basic reporting

The study entitled “Auditory ERP response to successive stimuli in infancy” has valuable information about infant’s neuronal myelination which is responsible to conduct hearing stimuli.

Here are some comments:
1. The manuscript is well written. But throughout the manuscript there are some mistakes like lack of space, lack of “full stop” etc. Please read carefully to correct those.
2. In “Introduction”, line 9 from below: Explain about the meaning of ‘N1, P1, N2, P2’clearly according to those mentioned references.
3. In ‘Procedure’.
a) Please mention the whole recording time of stimuli.
b) Mention also the size of net.
4. In ‘EEG analysis’.
a) Line 5 from below: Please mention “100 ms before” or (-100 ms) anyone for the segmentation.
b) Explain how movement and eye blink artifacts were removed?
5. The traces of “Figure 2” are not clear. All traces looks same and difficult to differentiate. Please clear this point.

Experimental design

Good.

Validity of the findings

The findings are validated.

Additional comments

No comments.

Reviewer 2 ·

Basic reporting

References need to be follow the PeerJ style

Experimental design

The authors need to be clear in the following sections:
1) Participant' s age range, e.g. 4:04-5:05, what does it mean? Is it 4.04 to 5.05 months?
2) Stimuli: what are the frequencies of F3, G3 and A#3 piano notes after synthesized in Nyquist, keeping the C4 at 440Hz
3) Procedure: during movie playing in front of infants, was the sound off or on?
4) EEG analysis: recording was done with 128 channels, 15 electrodes were removed for artefact from muscle, artefact rejection was done on 25 frontal channels, finally data was presented from FC (3 channels) or Fz site, point is that why many channels while target is a few and 25 frontal channels ,is it for template for artefact reduction from FC?

Validity of the findings

No Comments

Additional comments

The authors studied the developmental auditory ERP responses from infants using successive stimuli that is strong part of this article

·

Basic reporting

I have some concerns regarding the literature in the introduction and the conclusion: As the tones used in the paradigm were the same length as spoken syllables, I missed literature on the processing of sounds with the same temporal structure (for review see Giraud & Poeppel, 2012; Telkemeyer et al., 2009). The findings of those authors, that also contributed studies with infants could contribute to your conclusions. I also missed the findings of Jing & Benasisch (2006) on brain responses to tonal changes in the first two years of life. I missed in the introduction the reason why you choose the specific age group of 4-, 8-, and 12-months.
I
Where the participants from Dutch speaking environments? I had the idea, that Dutch children at the start of language acquisition are prone to concentrate on the first tone, as they also do concentrate on the first accentuated syllable in Dutch to find out where a new word in the speech stream starts. Literature on this can also contribute to your conclusion.
What is your explanation for the u-shaped waveform in the 8-month-olds?

Experimental design

In the method section, I wondered what led you to the participant number of 18? I also missed information about how many electrodes were interpolated and how long the experiment was in total.
Do I understand it right, that the infants heard 120 repetitions of the same melody? I would appreciate analysis where the ERPS are split by first vs. last presentation or grouping trials in different blocks so it can be seen if the three age groups show differences in repetition effects (decrease in amplitude) over the time course of the experiment.

Validity of the findings

no comments

Additional comments

The current study is looking at the development of auditory event-related potentials (ERPs) in infants. 4-, 8-, and 12-month-old ERPs to musical melody were obtained. For all notes, an overall reduction in latency for both P1 and N2 over age was found. N2 latency decreased significantly from 4 to 12 months. The most interesting finding are the different waveforms for the three notes in the three age groups. The 4-month-olds, the P1-N2 deflection was attenuated for the second and third notes, for the 8-month-olds, the attenuation was only found for the middle note; for the 12-month-olds, the P1 and N2 peaks show equivalent amplitudes. I found the paper well written and interesting to read.
I think the data set is relevant, the EEG/ERP analyses and the methods in general are good, and the study could be of interest to a developmental readership.

---

## Round 0.2 · accepted · Accept

Dear Authors, Congratulations!
Thank you for your revised manuscript which has received positive reviews thus the manuscript is accepted for publication.

·

Basic reporting

Good.

Experimental design

Good.

Validity of the findings

Good.

Additional comments

No comments.

·

Basic reporting

I am happy with the version of the manuscript now

Experimental design

none

Validity of the findings

none

Additional comments

none